# Insights into the Effect of Charges on Hydrogen Bonds

**DOI:** 10.3390/ijms25031613

**Published:** 2024-01-28

**Authors:** Andrea Chimarro-Contreras, Yomaira Lopez-Revelo, Jorge Cardenas-Gamboa, Thibault Terencio

**Affiliations:** 1School of Chemical Sciences and Engineering, Yachay Tech University, Urcuqui 100115, Ecuador; andrea.chimarro@yachaytech.edu.ec (A.C.-C.); jorge.cardenas@yachaytech.edu.ec (J.C.-G.); 2CATS Research Group, Yachay Tech University, Urcuqui 100115, Ecuador; 3School of Physical Sciences and Nanotechnology, Yachay Tech University, Urcuqui 100115, Ecuador; yomaira.lopez@yachaytech.edu.ec; 4Max Planck Institute for Chemical Physics of Solids, 01187 Dresden, Germany

**Keywords:** hydrogen bond, charge, spectroscopic variations, charge transfer, molecular overlap

## Abstract

Previous computational and experimental studies showed that charges located at the surroundings of hydrogen bonds can exert two opposite effects on them: rupture or strengthening of the hydrogen bond. This work aims to generalize the effect of charges in different hydrogen-bonded systems and to propose a coherent explanation of this effect. For these purposes, 19 systems with intra- and intermolecular hydrogen bonds were studied computationally with DFT. The FT-IR spectra of the systems were simulated, and two energy components of the hydrogen bond were studied separately to determine their variation upon the presence of a charge: charge transfer and molecular overlap. It was determined that either the breaking or strengthening of the hydrogen bond can be favored one over the other, for instance, depending on the heteroatom involved in the hydrogen bond. In addition, it is showed that the strengthening of the hydrogen bond by the presence of a charge is directly related to the decrease in charge transfer between the monomers, which is explained by an increase in molecular overlapping, suggesting a more covalent character of the interaction. The understanding of how hydrogen bonds are affected by charges is important, as it is a key towards a strategy to manipulate hydrogen bonds at convenience.

## 1. Introduction

Hydrogen bonds (HBs) are defined by IUPAC as an attractive interaction between a hydrogen atom from a molecule or a molecular fragment Y-H (donor), in which Y is more electronegative than H, and an atom or a group of atoms in the same or a different molecule, in which there is evidence of bond formation [1]. A typical hydrogen bond is depicted as Y-H⋯Y′-Z, where the dots denote the bond, Y-H represents the hydrogen bond donor, and Y’ the acceptor [1,2,3,4,5]. The atoms Y and Y′ usually correspond to heteroatoms in conventional HBs. However, non-conventional hydrogen bonds can have H donors such as C-H or H acceptors with π-bonded functional groups, halogens, or C atoms [6]. This is a special type of dipole–dipole interaction that is generally stronger than a normal dipole–dipole interaction but weaker than a covalent bond (10–30 kJ/mol) [5]. HBs are fundamental in a wide variety of systems; for instance, they play integral roles in biological structure, function, and conformational dynamics, so they are fundamental to life [4,7]. Moreover, they play a crucial role in material science, for instance, in the structure of hydrogels, in which they determine their physical and chemical properties [5]. In addition, they are relevant in novel technological applications such as batteries or organic semiconductor devices. For instance, the disruption of hydrogen bonds in the electrolyte enhances the stability of proton batteries [8]. Given their importance for structure and functionality in different systems, it would be useful to be able to manipulate them to convenience.

Even though HBs were thought to be simply electrostatic interactions, experimental observations (spectroscopic variations) suggested that other energetic contributions are present [9,10]. Moreover, it was determined that HB has a partially covalent character [11]. Also, a theoretical study on the energy components of the HB of a hydrogen-bonded water system determined that even though the major component is the electrostatic term, there are no negligible contributions of dispersion interactions as well as orbital interactions (polarization of electron density between the monomers) [12]. A model, including seven different energy components in HBs, has been proposed: electrostatic interactions, charge-transfer interactions, π-resonance assistance, steric repulsion, cooperative effects, dispersion interactions, and secondary electrostatic interactions [10]. Studies have shown the importance of these energy components in HBs. For instance, computational studies using SAPT showed that cooperativity effects [13] and π-resonance assistance [14] play important roles in the strengthening of HBs. Moreover, it was also determined computationally that charge transfer has an important role in HBs, with charge transfer interactions being responsible for 40% of the overall interaction energy within the water dimer [15].

It has been reported previously that charges have an effect on hydrogen bonds, which is determined experimentally as spectroscopic variations in X-ray absorption, X-ray diffraction, FT-IR, and Raman, caused by the interaction of cations of different nature with liquid water [16,17,18,19,20,21,22]. However, the reports on the interactions of HBs with charges are not perfectly congruent, displaying considerable variation both on the effect reported and its explanation. For instance, it was reported that both monovalent and divalent cations cause spectroscopic variations in X-ray absorption of liquid water, the first one mainly due to the anion, and the second one due to the cation showing variations depending on its identity [16]. These spectroscopic variations are attributed to a redistribution of charge among water molecules that solvate cations, causing charge transfer from the solvating water molecules to the cation [16]. The charge transfer then causes a perturbation of the unoccupied water molecular orbitals, and its magnitude depends on the identity of the cation (small charge transfer, less perturbation, and vice versa) [16]. Also, it was described that mono-, di-, and trivalent cations cause spectroscopic variations in X-ray diffraction of liquid water, showing that the hydrogen bond is being strengthened [22]. This hydrogen bond enhancement is attributed to the electrostatic influence of the cation on the hydrogen-bonded molecules, which polarizes them and changes the character of the HBs, increasing the bond order between Y″ and H (ionic charge on the oxygen increases, and electron population on hydrogen decreases) [17]. This increase in the bond order strengthens the HB. Moreover, it was determined that mono- and divalent cations cause spectroscopic variations in X-ray and SAXS of liquid water, showing that the hydrogen bond is being weakened in the first case and strengthened in the second [18].

In addition, the effect of a charge in systems different than water were also studied. For instance, it was determined that Li^+^ strongly enhances the donor–acceptor interaction of the hydrogen-bonded system HCONH_2_-OH_2_ [14]. Moreover, it was reported that the addition of Li^+^ to chitosan breaks the hydrogen-bonded network, which was determined by FT-IR spectroscopic variations [20]. It is noteworthy that theoretical studies on the effect of charges on HB have also been performed. For example, it was stated that cations can modify the Y-H length, the distance between heteroatoms, and the angle in a different manner, depending on the position with respect to the HB [21]. Also, it was determined that charges can exert effects on the HBs in DNA base pairs. In this case, metal cations can strengthen or weaken hydrogen bonds depending on the coordination position and on the atom with which it interacts (oxygen or nitrogen in the base pairs) [22,23]. Interactions with nitrogen weaken the bonds, whereas interactions with oxygen strengthens them (interactions with oxygen are stronger). This strengthening is explained by the stabilization of the accepting σ-LUMOs of the base pair due to the presence of the cation [22].

It is evident that charges do have an effect on HBs, and its explanation relies on the variation in the different energy components of the HBs. However, there is still a lack of generalization of this effect for different systems and a clear explanation of the nature of it. Therefore, the aim of this study is to propose a general description of the effect of the charges on different hydrogen-bonded systems and to present a coherent explanation of it. For these purposes, 19 molecules with intra- and intermolecular HB were studied computationally using DFT. Moreover, two energy components of the HB were studied separately to determine how they varied with the presence of the charge: molecular overlapping and charge transfer. It was determined that the HBs can be maintained (enhanced) or broken, and any of these effects can be favored over the other. Also, the HB strengthening is mainly explained due to a balance between the energy components of HB, in which molecular overlapping is favored, causing a decrease in charge transfer.

## 2. Results

### 2.1. Effect of the Charge on Hydrogen Bonds

Geometric and spectroscopic variations (FT-IR) of different hydrogen-bonded systems were studied in order to determine how they varied with the charge.

#### 2.1.1. Water and Ammonia

When the charge is located in the surroundings of the HB, two lowest-energy systems are obtained for water and ammonia. In the first case, the HB is maintained (Figure 1b and Figure 2b), and it is broken in the second (Figure 1c and Figure 2c). These systems will be designated as Case I and Case II, respectively. In both systems, Case II is more stable in energy than Case I by 53 and 94 kJ/mol, respectively.

Geometric variations due to the presence of the charge are observed in both systems with respect to the hydrogen-bonded system without charge (Figure 1a–c and Figure 2a–c). In Case II, the Y-H bond length decreases, and the distance between both heteroatoms increases. In Case I, the Y-H bond length increases, and the distance between both heteroatoms decreases. Moreover, the angle between the atoms involved in the HB increases towards 180°.

Spectroscopic variations in the FT-IR spectrum with respect to the hydrogen-bonded system without charge are also observed due to the presence of the charge (Figure 1 and Figure 2). Both the O-H and N-H stretching vibration of the acceptor and the donor are affected by the presence of the charge in its surroundings, showing a shift with respect to the corresponding stretching vibration of the hydrogen-bonded molecule without charge. This shift is more significant for the HB donor, so the stretching vibrations described in this paper refer to the Y-H stretching vibration. This vibration’s frequency decreases in Case I, and it increases in Case II, with respect to the hydrogen-bonded system.

The geometric and spectroscopic variations are summarized in Table 1, and they suggest that the HB is being strengthened in Case I. The strengthening and rupture of the HB (Case I and II) shows that the experimental results reported by [16,17,18,19,20,21,22] correspond to the two possible cases of the interaction between the charge and the HB.

#### 2.1.2. Other Systems

Table 1 summarizes the results of the study of 19 systems with intra- and intermolecular HB with a charge in its surroundings. The following considerations must be taken into account:The values of frequency, bond length, distance, and angle reported are the differences with respect to the hydrogen-bonded system without charge;Negative and positive values indicate decrease or increase with respect to the hydrogen-bonded system without charge, respectively;The energy difference reported is the difference of Case II with respect to Case I.

In the majority of the cases, the following behavior is observed: in Case II, the N-H or O-H bond lengths decrease, and the N-N or O-O distances increase with respect to the hydrogen-bonded system without charge; in Case I, these same distances increase and decrease, respectively. Moreover, in Case I, the angle increases towards 180°. Finally, in Case II, the O-H or N-H stretching vibrations increase with respect to the hydrogen-bonded system without charge; in Case I, these vibrations decrease. These observations suggest that the HB is being enhanced by the charge in Case I in all of the systems.

### 2.2. Nature of the Effect of the Charge on Hydrogen Bonds

Two energy components of the HB are studied in order to determine how they vary with the presence of a charge: charge transfer and molecular overlapping.

#### 2.2.1. Charge Transfer

Charge transfer of the 14 systems with intermolecular HB studied was determined by obtaining the difference between the Mulliken charges of the two molecules involved in the HB. In the majority of the cases, charge transfer decreases in Case I, and it is reduced to approximately 0 in Case II (Table 2). The charge transfer’s decrease in Case II is expected, given that the water molecules are no longer involved in a HB. On the contrary, the decrease in charge transfer in Case I might indicate that the molecular orbitals (MOs) involved in the HB are closer in energy than in a normal hydrogen-bonded system, decreasing the probability of charge transfer between them.

Moreover, a dependency of the charge transfer and the variation in the frequency of the Y-H stretching vibration in three areas is observed (Figure 3). The HB systems containing oxygen show higher variations in the frequency of the Y-H stretching vibration in comparison to the HB systems containing nitrogen. HB systems containing aromatic rings do not fall into any of these regions and can also show negative charge transfer values; this unexpected behavior might be caused by the electron delocalization of the aromatic rings. In addition, the results confirm the behavior reported by [16], in which a higher charge transfer causes more perturbations and, therefore, more significant spectroscopic variations.

For the five intramolecular systems studied, the Mulliken charges of the atoms involved in the hydrogen bond were analyzed (Table 3). In the majority of the systems in Case I, the Mulliken charge of the heteroatom of the acceptor becomes more negative, and the bridging hydrogen becomes more positive with respect to the hydrogen-bonded system. These results confirm the behavior reported by [17], in which the hydrogen bond enhancement is attributed to an increase in the bond order between Y″ and H (increasing the ionic charge of oxygen and decreasing that of hydrogen).

In addition, the charge transfer of water when hydrogen-bonded and in Case I was obtained at different levels of calculation containing different correlation percentage: from high correlation component (CC-SDT) to no correlation (HF) (Table 4) [24]. It was observed that the higher the level of calculation the more the positive charge of cation is transferred to the HB donor and acceptor (in more proportion to the donor). This also seems to vary the charge transfer between the donor and the acceptor, with respect to the hydrogen-bonded system without charge at all levels of calculation. Moreover, the decrease in the charge transfer in the CC-SDT and MP2 methods (in which the dynamic correlation is considered with more accuracy) are very similar. On the contrary, in the CAS and HF methods, in which only a part of the dynamic correlation or no correlation at all are considered, the values vary considerably, and the charge transfer’s behavior is different. Therefore, the correlation component, and more specifically the dynamic one, is important to explain the variation in the charge transfer of hydrogen-bonded systems with a charge in its surroundings.

It is noteworthy that DFT describes this behavior well, but it overestimates it, which is expected due to the self-interaction error of DFT. Consequently, DFT can be used to explain the effect of charges on HBs qualitatively (as was performed in this paper), but the addition of the correlation component must be taken into account in order to have a more quantitative description.

#### 2.2.2. Orbitals Involved in HB

The orbitals involved in the HB were studied in DFT in the hydrogen-bonded water system and in Case I. In the hydrogen-bonded system without charge, there are two MOs that show direct orbital interactions between the monomeric orbitals, favoring the HB (Figure 4b,c). On the contrary, in Case I, there are three MOs that show direct orbital interactions between the two monomers (Figure 5a–c) Moreover, the orbital HOMO-6 of the hydrogen-bonded system without charge is only located on the acceptor molecule (Figure 4a), but the addition of a charge causes this orbital to become delocalized in both monomers, favoring the HB.

### 2.3. Evaluation of the Charge Transfer

The charge transfer of the systems alone and in Case I is studied using the following formalisms: CHELGP, AIM, NBO, Mulliken, and Löwdin. Different tendencies regarding the variations in charge transfer with the presence of the charge are observed depending on the formalisms used (Table 5). For CHELGP, AIM, and NBO, an increase in charge transfer is observed when the charge is near the HB. On the contrary, a decrease in the charge transfer with the charge near the HB is obtained using Löwdin and Mulliken.

## 3. Discussion

In all of the systems studied, whether they have intra- or intermolecular HB, two lowest-energy structures were obtained: the case in which the HB is maintained (Case I), and the case in which it is broken (Case II). The difference in energy between both favors Case II over Case I in all of the systems and varies between 32.2 and 97.73 kJ/mol. This suggests that is more probable that the addition of a charge to the system favors the HBs’ rupture (Case II), which is observed experimentally in previous studies [18,19,20]. However, given the fact that the enhancement of HBs with the addition of a charge was also previously reported [17,19,21,22], Case I is also observed experimentally. Therefore, this indicates that one case can be favored over the other. For instance, it is suggested that the weakening of the HB (Case II) is more favored than its enhancement (Case I) for HBs involving nitrogen rather than oxygen, due to the higher difference in energy between both cases (89–94 kJ/mol). This confirms the behavior reported by [17]. Moreover, the variation in the frequency of the Y-H stretching vibration is higher for HBs involving oxygen, so it seems that charges favor the enhancement of HBs containing oxygen. However, other factors could also favor a case over the other, for instance, the cation concentration, the identity of the contra-anion, or the presence of more complex structures that favor the attraction of the charge.

Case II favors the rupture of the hydrogen bond, causing geometric and spectroscopic changes with respect to the hydrogen-bonded system without charge. On one hand, the decrease in the Y-H bond-length and the increase in the distance between both heteroatoms involved in the HB are observed. On the other hand, an increase in the frequency of the Y-H stretching vibration is observed. This increase in the vibration frequency is expected, given that free molecules that are not involved in HBs show higher frequencies than hydrogen-bonded molecules, which suggests that this bond is stronger [20,21]. Then, as this bond is stronger, its bond distance is smaller too [1,25]. Moreover, the fact that the charge transfer between both molecules is approximately zero indicates that there is no longer interaction between them. Case II could be useful for systems, in which it is due to weakening intermolecular interactions. For instance, the weakening of hydrogen bonds in lithium-ion batteries (LIBs) with organic electrodes such as indanthrone (IDT) enhances active component utilization, resulting in a nitrate-activated IDT cathode with a prominent voltage plateau (2.5 V) and initial discharge capacity of 226 mAh g^−1^ at 0.1 C, nearing its theoretical capacity [26].

Case I enhances the HB, causing geometric and spectroscopic changes with respect to the hydrogen-bonded system without charge. On one hand, the Y-H bond-length increases, and the distance between both heteroatoms involved in the HB decreases. In addition, the angle between the atoms involved in the HB increases. On the other hand, the frequency of the Y-H vibration decreases. The decrease in the frequency of this vibration suggests that this bond is weakened [27,28], so the bond distance is expected to be higher [1,25]. This weakening suggests that this bond is more involved in the intermolecular interaction, which means that the HB is being strengthened. Moreover, the decrease in the distance between both heteroatoms involved in the HB and the increase in the angle towards 180° confirm the strengthening [22]. Case I could be useful for systems with poor intermolecular interactions that need to be enhanced. For example, it has been found that strengthening hydrogen bonds in graphene oxide (GO) sheets and their associated polymeric nanocomposites enhances their mechanical properties. In particular, stiffness increases continually to a maximum value of 60 GPa when 7 wt % water is present [29].

The enhancement of the HB (Case I) can be explained by the change in the contributions of the different energy contributions of the HB by the presence of a charge. Some of the energy contributions are distance dependent; for instance, the further the monomers are, the more the electrostatic interaction term increases, and the closer they are, the more orbital overlap is observed [13]. As showed previously, the presence of the charge decreases the distance between heteroatoms, so the monomers are closer. Given the decrease in the distance, the molecular overlap must be favored over electrostatic interactions, strengthening the bond. The increase in molecular overlap is confirmed with the MO analysis, which shows that there are more MOs interacting directly when the charge is present, suggesting that they are closer in energy than in the normal hydrogen-bonded system. The correlation component favors the transfer of positive charge from the cation to the donor molecule, so it is suggested that the charge affects the energy of the MOs centered in the donor. Moreover, due to the fact that the MOs of the acceptor are lower in energy than that of the donor, the charge will lower the donor MOs’ energy. Finally, due to the fact that the orbitals are closer in energy, the charge transfer between the molecules is decreased. A model scheme of this behavior is shown in Figure 6.

The orbital model proposed (Figure 6) is coherent with Mulliken charges, corresponding to atomic orbital equilibration and weakening of charge transfer, respectively, when introducing Li^+^. However, Mulliken charges are, in general, known to be imprecise, sensible to the systems (particularly inorganic), and basis-set dependent [30,31]. Therefore, the variation in charge transfer due to the presence of the charge was studied with different charge calculation methods: orbital wavefunction methods (Mulliken, Löwdin and NBO), electron density partition methods (AIM), and electrostatic potential fitting methods (CHELPG). Due to the fact that atomic charges cannot unambiguously be determined by quantum chemical calculations, these vary significantly depending on the formalism used [30,31,32]. The model proposed in this paper regarding the equilibration of the charge transfer when the cation is introduced is in contradiction with the results obtained with CHELPG, AIM, and NBO. Even though it could be expected that formalisms other than orbital-based methods would give more accurate results, it was reported that methods such as CHELPG are not necessarily the most suitable for non-covalent interactions cases [31]. Therefore, this work suggests that charges derived from the wave functions, such as Mulliken and Löwdin, are better descriptors of how the charge transfer is affected in the presence of a charge, whereas the other formalisms would provide a contrary behavior and predict an increase in charge transfer.

## 4. Materials and Methods

The following 19 systems containing intra- or intermolecular hydrogen bonds were studied: water, ammonia, methylamine, methanol, ethanol, phenol, 4-hydroxybutanoic acid, ethylene glycol, 4-nitrophenol, 2-hydroxybenzaldehyde, 4-(phenyldiazenyl)phenol, hydroxyaniline, cyclohexanol, propylene glycol, cyclopropanol, 1,2-benzenediol, butanol, phenylmethanamine, and ethylamine. Lithium cation was used to simulate a charge model. The optimization of geometry and the infrared spectra of the systems with and without charge were obtained with the software orca 4.1.0 [33] with the following parameters: B3LYP functional, def2-TZVP basis set [34], and the dispersion correction D3BJ [35]. In addition, the hydrogen-bonded water system and Case I were studied at the following levels of theory: HF, MP2, DFT, CCSD, and CAS. The basis set cc-pVTZ [36,37] was used for all of the cases and the auxiliary basis sets def2-SVP/C [38] and cc-pVTZ/C [39,40] were used for CAS and MP2 respectively. All these were SP calculations using the cc-pVTZ basis-set with auxiliary basis sets. Finally, different charge calculation methods were used to obtain the charge transfer: Löwdin in the software orca, CHELPG and AIM in the software Multiwfn [41,42], and NBO in the software JANPA [43,44].

## 5. Conclusions

Charges can exert an effect on HBs: whether they break the bond or strengthen it. Both cases can be identified through specific geometric and spectroscopic variations, and it seems that Case I is more favored in HBs involving oxygen, and Case II is more favored in HBs with nitrogen involved. Other factors could also favor one case over the other, such as the nature of the contra-anion or the cation’s concentration. In addition, the strengthening of the HBs by the presence of a charge seems to be related to the repartition of the different energy components of HBs, favoring molecular overlap over electrostatic interactions, strengthening the bond.

This investigation proposes a simple model to explain the effect of charges on HBs: the transfer of positive charge from the cation to the donor molecule causes a change in the energy of the MOs centered on the donor part, lowering their energy and thus resulting in energy closer to that of the acceptor. The similarity in energy of the MOs of the acceptor and the donor decreases the charge transfer between the monomers, as observed from more symmetric orbitals and Mulliken or Lowdin population analysis. Moreover, this transfer of positive charge towards the donor is favored by the correlation component.

With further investigation on this matter, it should become possible to be able to manipulate hydrogen bonds at exact convenience.

## Figures and Tables

**Figure 1 ijms-25-01613-f001:**
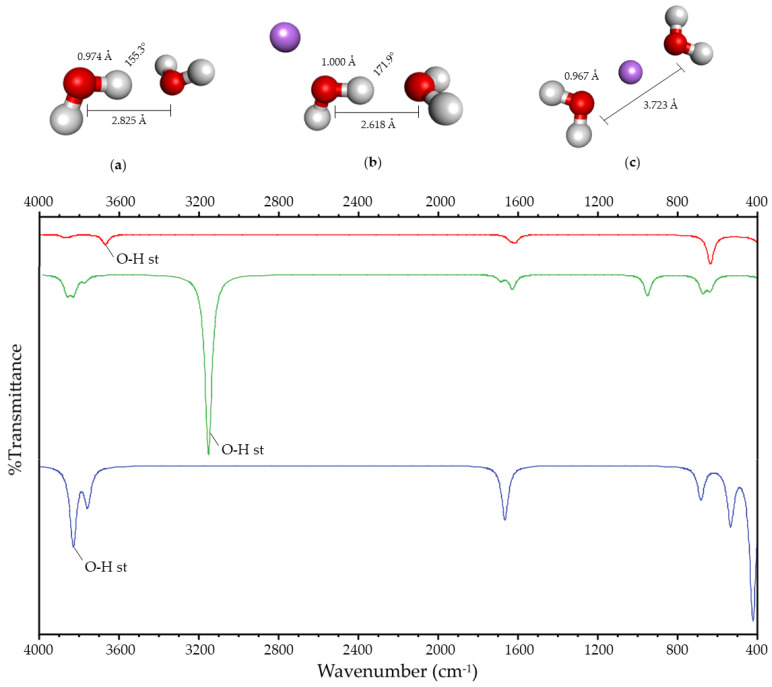
(**Up**): Optimized geometry of (**a**) hydrogen-bonded water; (**b**) Case I; and (**c**) Case II. Atom colors are shown as follows: oxygen in red, hydrogen in gray, and lithium ion in purple. The values shown correspond to the O-H bond length, the O-O distance, and the O-H-O angle. (**Down**): Simulated infrared spectra of hydrogen-bonded water (in red); Case I (in green); and Case II (in blue).

**Figure 2 ijms-25-01613-f002:**
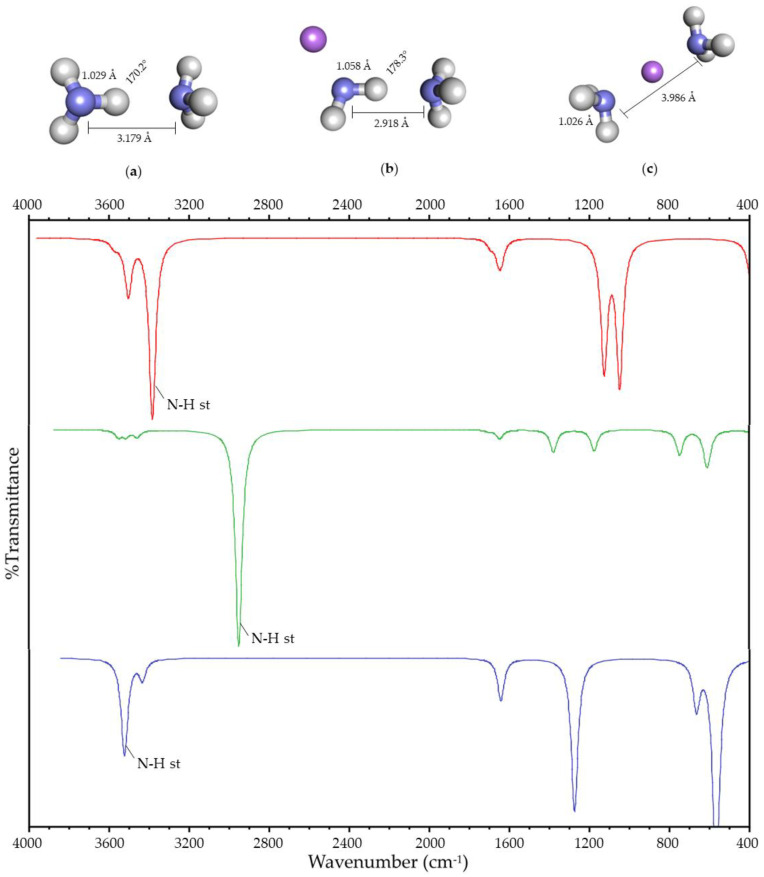
(**Up**): Optimized geometry of (**a**) hydrogen-bonded ammonia; (**b**) Case I; (**c**) and Case II. Atom colors are shown as follows: nitrogen in blue, hydrogen in gray, and lithium ion in purple. The values shown correspond to the N-H bond length, the N-N distance, and the N-H-N angle. (**Down**): Simulated infrared spectra of hydrogen-bonded ammonia (in red); Case I (in green); and Case II (in blue).

**Figure 3 ijms-25-01613-f003:**
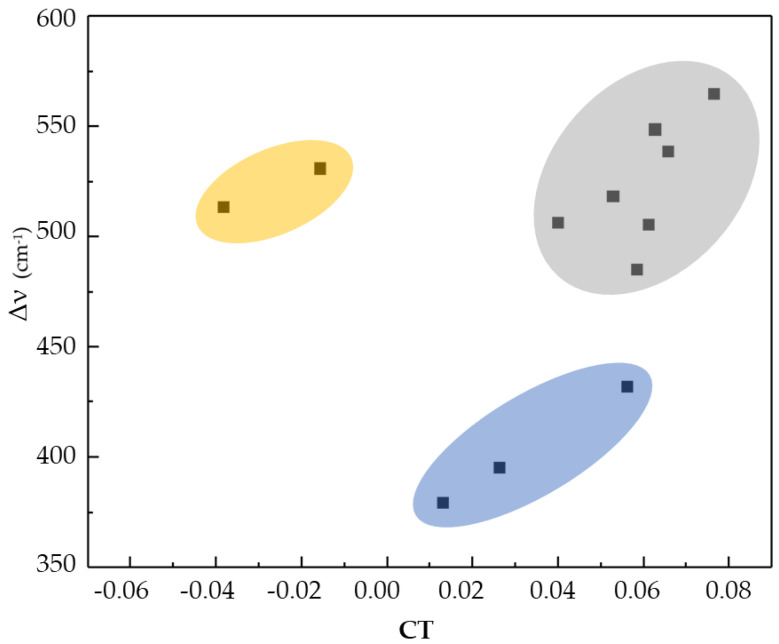
Charge transfer vs. variation in Y-H stretching vibration frequency. In blue: N-H stretching vibration; in gray: O-H stretching vibration; and in yellow: HB systems containing aromatic rings.

**Figure 4 ijms-25-01613-f004:**
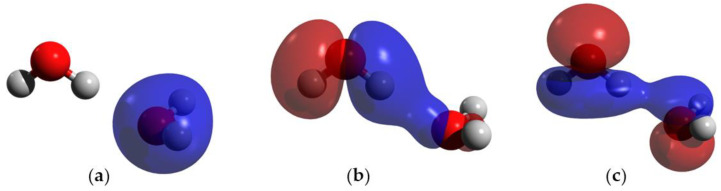
MOs of the hydrogen-bonded water system; blue and red indicate opposite signs of the molecular orbital. (**a**) HOMO-6; (**b**) HOMO-4; and (**c**) HOMO-2. Atom colors are shown as follows: oxygen in red, and hydrogen in gray.

**Figure 5 ijms-25-01613-f005:**
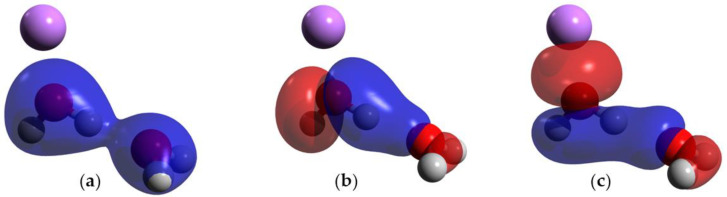
MOs of Case I; blue and red indicate opposite signs of the molecular orbital. (**a**) HOMO-7; (**b**) HOMO-5; and (**c**) HOMO-3. Atom colors are shown as follows: oxygen in red, hydrogen in gray and lithium ion in purple.

**Figure 6 ijms-25-01613-f006:**
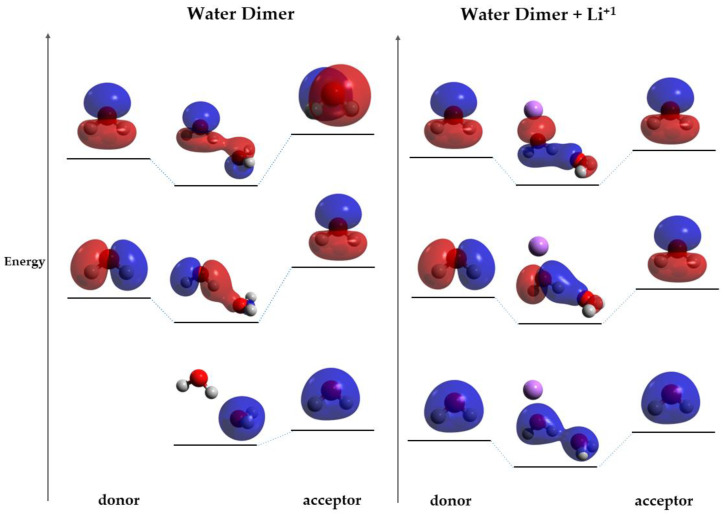
Comparison between the MOs involved in the HB of the hydrogen-bonded system (**left**) and Case I (**right**); blue and red indicate opposite signs of the molecular orbital. Orbital energy levels are represented as solid bars. The bars on the left and right sides correspond to the orbitals of the donor and acceptor monomer respectively. The dotted lines indicate the monomer orbital that contributes to the molecular orbital. Atom colors are shown as follows: oxygen in red, hydrogen in gray and lithium ion in purple.

**Table 1 ijms-25-01613-t001:** Summary of geometric and frequency description of several hydrogen-bonded systems with and without charge.

System	Case I				Case II			Energy Difference (kJ/mol)
	Frequency (cm^−1^)	^a^ Y-H Length (Å)	^a^ Y-Y″ Distance (Å)	Angle (°)	Frequency (cm^−1^)	^a^ Y-H length (Å)	^a^ Y-Y″ Distance (Å)	
Water	−518.5	0.026	−0.207	16.6	159.3	−0.007	0.898	53
Ammonia	−432.27	0.029	−0.261	8.1	141.69	−0.003	0.807	93.7
Methylamine	−395.6	0.015	−0.172	4.6	27.6	0.074	0.826	89.6
Methanol	−548.92	0.031	−0.246	11.6	146.27	−0.005	0.837	59.5
Ethanol	−565.02	0.025	−0.205	12.5	154.28	−0.011	0.870	65.4
Phenol	−531.18	0.028	−0.233	4.7	139.21	−0.006	0.752	32.2
Ethylene glycol	−538.9	0.028	−0.227	10.7	118.6	−0.006	0.445	75
4-nitrophenol	−513.7	0.026	−0.224	13.3	122.2	−0.006	0.537	57
4-(Phenyldiazenyl)phenol	−353.7	0.018	−0.121	−2.7	149.5	−0.007	0.670	70,8
^b^ 4-Hydroxybutanoic acid	−845.34	0.044	−0.385	37.8	43.34	0.002	0.039	66
^b^ 2-Hydroxybenzaldehyde	−1190.5	0.077	−0.192	9.6	412.6	−0.02	0.143	64.9
^b^ Hydroxyaniline	−609.8	0.036	−0.157	12	153.9	−0.008	−0.005	62.7
Cyclohexanol	−505.60	0.026	−0.257	0.9	165.83	−0.008	0.771	74.3
^b^ Propylene glycol	−270.41	0.012	−0.194	11	42.27	−0.001	0.34	−24.1
Cyclopropanol	−485.39	0.032	−0.280	5.8	146.14	−0.007	0.807	65.2
^b^ 1,2-Benzenediol	−103.34	0.006	−0.067	0.5	29.49	−0.002	−0.101	64.3
Butanol	−506.55	0.036	−0.285	1.8	167.48	−0.007	0.79	71.8
Phenylmethanamine	−361.76	0.014	−0.194	2.3	20.70	−0.004	0.73	94.1
Ethylamine	−379.46	0.024	−0.227	11.7	36.60	−0.001	0.769	94.4

^a^ can correspond to nitrogen or oxygen depending on the system. ^b^ Intramolecular hydrogen-bonded systems.

**Table 2 ijms-25-01613-t002:** Charge transfer of several intermolecular hydrogen-bonded systems with and without charge.

System	Hydrogen Bond	Case I	Case II
Water	0.1424	0.0527	0.0003
Ammonia	0.1684	0.0560	0.0003
Methylamine	0.1465	0.0262	0.0031
Methanol	0.0596	0.0625	0.0001
Ethanol	0.1119	0.0763	0.0134
Phenol	0.0739	−0.0159	0.0213
Ethylene glycol	0.0397	0.0656	0.0314
4-nitrophenol	0.0294	−0.0384	0.0066
4-(Phenyldiazenyl)phenol	0.0336	0.0228	0.0058
Cyclohexanol	0.0448	0.0610	0.0205
Cyclopropanol	0.0573	0.0583	0.0029
Butanol	0.0446	0.0398	0.0007
Phenylmethanamine	0.0811	−0.0527	0.0026
Ethylamine	0.0119	0.0129	0.0007

**Table 3 ijms-25-01613-t003:** Mulliken charges of several intramolecular hydrogen-bonded systems with and without charges.

System	Case	* Y	H	* Y″
4-Hydroxybutanoic acid	Hydrogen bond	−0.455178	0.317549	−0.328793
	Case I	−0.693745	0.391354	−0.358354
	Case II	−0.39869	0.230582	−0.339584
2-Hydroxybenzaldehyde	Hydrogen bond	−0.366896	0.341047	−0.356051
	Case I	−0.580469	0.373013	−0.351110
	Case II	−0.340888	0.300074	−0.439170
Hydroxyaniline	Hydrogen bond	−0.422056	0.310226	−0.504710
	Case I	−0.643747	0.364912	−0.576466
	Case II	−0.561430	0.366445	−0.741082
Propylene glycol	Hydrogen bond	−0.4704	0.30660	−0.4785
	Case I	−0.6702	0.3737	−0.4886
	Case II	−0.6171	0.3713	−0.4043
1,2-Benzenediol	Hydrogen bond	−0.4040	0.3257	−0.4559
	Case I	−0.6352	0.3811	−0.4560
	Case II	−0.5858	0.3613	−0.5858

* can correspond to nitrogen or oxygen, depending on the system.

**Table 4 ijms-25-01613-t004:** Charge transfer of water with and without charge at different levels of theory.

Level of Theory	HB		Case I		
	Donor	Acceptor	Donor	Acceptor	Cation
CC-SDT	−0.0409	0.0410	0.1803	0.1124	0.7072
MP2	−0.0426	0.0426	0.1824	0.1152	0.7023
CAS	−0.0355	0.0355	0.1756	0.0924	0.7321
HF	−0.0297	0.0298	0.1600	0.0898	0.7502
DFT	−0.0496	0.0495	0.2129	0.1268	0.6604

**Table 5 ijms-25-01613-t005:** Variations in the charge transfers of several intermolecular hydrogen-bonded systems after additions of charges. Negative and positive values indicate decrease or increase with respect to the hydrogen-bonded system without charge, respectively.

System	Löwdin	Mulliken	CHELPG	AIM	NBO
Water	−0.0564	−0.0897	0.0171	0.0236	0.0707
Ammonia	−0.0917	−0.1124	0.0073	0.0350	0.0677
Methylamine	−0.129	−0.1203	0.0381	0.0147	0.0442
Methanol	−0.0454	0.0029	0.0307	0.0293	0.0772
Ethanol	−0.0888	−0.0356	0.0010	0.0212	0.0670
Phenol	−0.119	−0.0898	−0.0257	0.0258	0.0670
Ethylene glycol	−0.0465	0.0259	0.0918	0.0387	0.0843
4-nitrophenol	−0.068	−0.0678	0.0554	0.0432	0.0790
4-(Phenyldiazenyl)phenol	−0.1423	−0.0108	0.0071	0.0156	0.0404
Benzylamine	−0.2855	−0.1339	−0.064	0.038	0.0146
Cyclohexanol	−0.1064	0.0162	−0.1360	0.026	0.0688
Butanol	−0.1341	−0.0048	0.0096	0.0179	0.0553
Cyclopropanol	−0.1052	0.0010	0.0162	0.0271	0.0657
Ethylamine	−0.1160	0.0009	0.1139	0.038	0.0616

## Data Availability

Not applicable.

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
