# Peer review of "Insights into the Effect of Charges on Hydrogen Bonds"

_ijms, 2024, doi:10.3390/ijms25031613_

Round 1
Reviewer 1 Report
The authors present a theoretical study of intermolecualr systems formed through the hydrogen bond phenomenology. Directly, the work can and should improve to reach the acceptability for publication, and the reasons for this statement can be easily and shortly summarized as follows:
Major remarks
- In the abstract, the first and second sentences (for both of them up to “...are not perfectly congruente.”) must be transferred to the introduction and followed by specific references in H-Bond studies, which are numerous and can be easily found in the literature. In my point of view, abstract should show a simplication of the research instead of definitions of the main theme. This same point of view can be also used in the final of the abstract, in particular, from:” It was determined that when Hydrogen Bonds are 18 strengthened, molecular overlapping is favored, so, the charge transfer between the monomers de- 19 crease. The understanding of how Hydrogen Bonds are affected by charges is important as it is a 20 key towards a strategy to manipulate Hydrogen Bonds at convenience.”;
- Since the authors are using charge transfer, wherein only the Mulliken approach was used, this is very limited because there are many other atomic charge methods, such as NBO, ChElPG, QTAIM, GAPT, and so on. The authors could explore all these ones formalisms, or even many other ones, in order to improve the quality of the discussion along the text. In view of this, it is widely known that results derived from Mulliken’ atomic charges are not consistentes as well as GAPT early cited, and regarding this last one, it can be useful for infrared absorption modes but when applied for intermolecular systems whose structures are linear and symmetrics;
- The use of Hartree-Fock results for H-Bond studies is useless;
- The linear trend presented in Figure 3 is strengthless, mainly for those systems with negative CT values. I would suggest to use more systems and since the propose is to obtain systematic tendencies, 12 sytems is a very limited number;
- In a couple years ago, the H-Bond studies at the light of the examining of decrease and increase of electronic density on proton donating and accepting centers has the main focus to evaluate the interaction strenght, and as such, I recommend to the authors the SAPT calculations, by which, the contribution of the charge transfer in this current work will be quantified. I have a brief idea of the interaction strength of the systems in this current work, even so the results (with BSSE na ZPE corrections) were not presented, and by taking into account the DFT, MPS and CCSD level of theory used, at least a slight covalency level may be obtained via SAPT runs and this can be extremely importante to contribute with the purpose of this work. I would like to emphasize that, all these comments are nor criticismo in essence, but necessaries suggestions to allign the work with all other ones of the intermolecular specialized literature.
Minor remarks
- The number of recently published references is very few, i.e., only two papers published in last three years (2020 and 2022) were cited. This is not usual due to the fact of the large set of references in hydrogen bond studies available nowadays.
Author Response
Dear Reviewer,
We deeply thank you for all your suggestions as it resulted in very interesting discussion (especially the comparison of different charge schemes) and we feel resulted in a real improvment of the manuscript. We provide in the attached file an answer to each point that you mention.We also made some additional small changes in order to explain our work more clearly and provided additional litterature.

Reviewer 2 Report
Hydrogen bonds of all kinds continue to be of great interest in both life and material sciences. Therefore, the effects of the environment (such as charge, which is studied in the present work MS) on the strength and geometry of hydrogen bonds are of great interest. However, many of these effects have already been studied in detail on more realistic models and the influences are more or less known. Nevertheless, some new aspects of these topics can be presented, and this was my aim when I started reading MS.
However, I very much regret that the present MS is far from providing new insights into the field of hydrogen bonding. The MS is unsystematically and confusingly written and lacks consistency. The model systems used are greatly simplified to provide new insights into the energetics and geometry of hydrogen bonds. Therefore, I have to reject MS for publication.
The inconsistent use of capital letters and abbreviations.
The writing style is confusing, using words that I really do not understand. For example spectroscopic variations
Author Response
Dear reviewer, we thank you for your time.
We try to change our manuscript to make it more understandable, also reviewing the english language more carefully.
We feel you were not totally convinced by the novelty of our work, so we try to support it more in the manuscript and also in the attached response.
We think that the most interesting point is the description of the change in energy level of the orbitals when the molecules are involved in H-bonds.
We try to explain and support our idea further.

Reviewer 3 Report
Review Report:
In their manuscript Chimarro et al. provided theoretical insights on the charge-effect on hydrogen bonding (HB) interactions. They showed how the positive charge in the surroundings affects the HB and other molecular parameters such as bond length, bond angle, charge transfer, and vibrational frequencies.
The work is important in the context of understanding this important noncovalent interaction in different conditions. Although the results are thorough, consistent, and fit with the journal scope, I expect some further discussions on certain aspects. I, therefore, recommend publication after addressing the following comments. My remarks are copied below. I use the following abbreviations, P-page number and L-line number.
Comments:
1. P1-L25: I encourage the author to provide the IUPAC definition of HB and provide the latest reference.
2. P1-L28: The authors should discuss the unconventional HB interactions in detail in the introduction section.
3. P3-Figure 1: The authors should mention the source of the FTIR spectrum.
The y-axis should be named.
The same comments should be followed for Figure 2.
4. P5-Table 1: The authors should clearly mention that the numbers are compared to the corresponding neutral counterparts.
5. P6-Table 2: There should units for the CT.
6. P6-Figure3: The axis units must be provided.
7. For the charge transfer, the authors mainly discuss the Mulliken charge. However, it has been questioned at times. Therefore, I encourage the authors to include the natural charges as well in their discussion.
Author Response
Dear reviewer,
we thank you very much for your suggestions and followed all recomendations to include it in the manuscript. The addition of other charges formalism is particularly interesting.
Please find our answers point by point in the attached file

Round 2
Reviewer 2 Report
As I stated in the first review, I still think that the biggest weakness of MS is the oversimplification of the model systems, which can lead to wrong conclusions. The calculation of charges is a very sophisticated process where several different methods can be applied. A very good example is described in reference Liu, X.; C. Heath Turner, Computational Study of the Electrostatic Potential and Charges of Multivalent Ionic Liquid Molecules. Journal of Molecular Liquids 2021, 340, 117190–117190, you used it in your MS. This is the right approach to solve the problem.
However, if we omit this point, the importance of the matter presented in MS, i.e. the influence of the charge on the HB strength and geometry, can override the weakness of the methodology and serve as a simple methodological approach to clarify the effect of the charges on the HB properties.
The English can be improved
Reviewer 3 Report
I am satisfied with the changes made by the authors. Thus, there is no further review required. I recommend publication of the manuscript in its present form.